# The Impact of Probiotics on Intestinal Mucositis during Chemotherapy for Colorectal Cancer: A Comprehensive Review of Animal Studies

**DOI:** 10.3390/ijms22179347

**Published:** 2021-08-28

**Authors:** Povilas Miknevicius, Ruta Zulpaite, Bettina Leber, Kestutis Strupas, Philipp Stiegler, Peter Schemmer

**Affiliations:** 1General, Visceral and Transplant Surgery, Department of Surgery, Medical University of Graz, Auenbruggerpl. 2, 8036 Graz, Austria; povilui.m@gmail.com (P.M.); ruta.zulp@gmail.com (R.Z.); bettina.leber@medunigraz.at (B.L.); peter.schemmer@medunigraz.at (P.S.); 2Faculty of Medicine, Vilnius University, 03101 Vilnius, Lithuania; kestutis.strupas@santa.lt

**Keywords:** gut microbiota, dysbiosis, proinflammatory cytokines

## Abstract

Colorectal cancer (CRC) is the second most commonly diagnosed cancer in females (incidence 16.4/10,000) and the third in males (incidence 23.4/10,000) worldwide. Surgery, chemotherapy (CTx), radiation therapy (RTx), or a combined treatment of those are the current treatment modalities for primary CRC. Chemotherapeutic drug-induced gastrointestinal (GIT) toxicity mainly presents as mucositis and diarrhea. Preclinical studies revealed that probiotic supplementation helps prevent CTx-induced side effects by reducing oxidative stress and proinflammatory cytokine production and promoting crypt cell proliferation. Moreover, probiotics showed significant results in preventing the loss of body weight (BW) and reducing diarrhea. However, further clinical studies are needed to elucidate the exact doses and most promising combination of strains to reduce or prevent chemotherapy-induced side effects. The aim of this review is to overview currently available literature on the impact of probiotics on CTx-induced side effects in animal studies concerning CRC treatment and discuss the potential mechanisms based on experimental studies’ outcomes.

## 1. Introduction

Colorectal cancer incidence and mortality rates vary markedly around the world. According to the World Health Organization (WHO), CRC is the third most commonly diagnosed cancer in males (incidence 23.4/10,000) and the second in females (incidence 16.2/10,000) globally, with 1.8 million new cases and almost 861,000 deaths in 2020 [1]. Rates are substantially higher in males than in females. The incidence of CRC is associated with modifiable and nonmodifiable risk factors. Genetic factors, gender, age, and ethnicity comprise nonmodifiable risk factors [2,3], whereas modifiable risk factors include low level of physical activity, diet and excess BW, lifestyle, inflammation, prescription drugs, smoking, alcoholic beverages, and, as recently reported, dysbiosis in the gut [4,5,6,7].

Chemotherapy, radiation therapy, and surgery are the primary treatment modalities for different cancer types, including CRC [8]. It has been proved that the efficacy of CTx alone or in combination with RTx is the foundation for treating cancer patients, including CRC patients [9]. However, CTx-induced intestinal mucositis (IM) is a major oncological problem that has been reported in 50–80% of patients, significantly affecting patients’ quality of life [10,11]. Inflammation usually accompanies cell loss in the epithelial barrier lining the gastrointestinal tract. Clinical symptoms of IM usually include nausea, bloating, vomiting, constipation or diarrhea, and weight loss. Moreover, mucositis frequently leads to dose reduction of CTx agents or even postponement, resulting in higher mortality [12,13]. Due to preclinical trials, major progress has been made in understanding the pathophysiological mechanisms of IM [14,15]. 

Some studies found preventive and therapeutic capacities of specific probiotic strains in different diseases such as infectious diseases, antibiotic- or CTx-induced diarrhea, liver insufficiency, lactose intolerance, inflammatory bowel disease, irritable bowel syndrome, and cancer [16,17,18,19]. Probiotics have been shown to confer beneficial effects on CRC and CTx-induced side effects because of the direct exposure of the colon to the consumed bacteria [20,21]. Moreover, it has been reported that probiotics have different abilities such as promoting crypt cell proliferation, preventing cytokine-induced apoptosis, reducing proinflammatory cytokine production, and regulating the intestinal immune system [22,23]. 

The objective of this comprehensive review is to provide an overview of the currently available literature on the impact of probiotics on CTx-induced side effects in animal studies concerning CRC treatment and discuss the potential mechanisms based on experimental studies’ outcomes. 

## 2. Materials and Methods

The literature search was performed in the PubMed, Web of Science, EMBASE, and clinicalTrials.gov online databases. The following combination of Medical Subject Headings (MeSH) and keywords with the employment of “AND” or “OR” Boolean operators were used: “Preclinical trials” OR “Gastrointestinal mucositis” OR “Chemotherapy side effects” OR “Cancer” OR “Diarrhea” OR “Colorectal cancer” OR “Intestinal microbiota” OR “Chemotherapy induced mucositis” OR “Chemotherapy induced diarrhea” OR “Chemotherapy” OR “Gut microbiota” AND “Probiotics”.

The search was restricted to English language only without a time limitation. Most recent search was performed on 17 May 2021.

At least two researchers reviewed the abstracts for the inclusion. After relevant abstracts were identified, full-text articles were retrieved and re-reviewed. Reference lists from selected studies were examined, and relevant articles included. 

## 3. Comprehensive Review

### 3.1. Probiotics

Probiotics are bacteria with health benefits ingested as a supplement or food constituent that have been consumed increasingly in recent years [24]. The Food and Agriculture Organization (FAO) and World Health Organization (WHO) have defined probiotics as “live microorganisms that, when administered in adequate amounts confer a health benefit on the host” [25]. According to the current definition, the term probiotics implies alive, viable bacteria; it does not apply to dead bacterial components. Moreover, probiotics should have several certain characteristics to exert maximum therapeutic effects, including resistance to the gastrointestinal tract environment (low pH and bile salt), because bacteria must remain viable, able to adhere to the intestinal mucosa, and able to colonize the intestinal tract [26]. There are many different microorganisms currently used as probiotics, with the most common group of probiotics belonging to the lactic acid bacteria of the genera *Lactobacillus* and *Bifidobacterium* (Table 1) [27]. 

Compared to pathogenic bacteria, probiotics are considered safe, and infections caused by probiotics are extremely rare. Probiotics are noninvasive despite strong adherence to the intestinal epithelium. Most studies did not report a statistically significant increase in adverse events compared to control groups [28,29]. Usually, probiotic bacteria colonize the intestine only transiently without producing toxins or metabolites dangerous for the host [30]. However, there are also clinical trials from which deaths were reported [31].

Probiotics are used to improve the homeostasis of internal microbiota in order maintain intestinal health [32]. As a result, the number of harmful bacteria that cannot survive in the acidic environment decreases while the beneficial bacteria that thrive in the acidic environment proliferate, balancing the intestinal microbiota [33].

Scientific evidence supports the important role that probiotics can play in the digestive system, having significant effects in alleviating the symptoms of GIT diseases such as: irritable bowel syndrome, inflammatory bowel disease, GIT infections, constipation, food allergies, antibiotic- or CTx-induced diarrhea, and colorectal cancer [21,34]. The orally administered probiotic cocktail VSL#3 has been shown to be effective in inducing remission in patients with mild to moderate ulcerative colitis by decreasing expression of TLR-4, NF-κB, and inducible nitric oxide synthase [35,36]. In pouchitis, it appears to exert several anti-inflammatory mechanisms of action, including alteration of cytokine profile and expression of nitric oxide synthase and matrix metalloproteinases [37]. Moreover, *Saccharomyces boulardii* prevented relapse from active disease in patients with Crohn’s disease and infections caused by Clostridium difficile [38]. Probiotics exhibited antiproliferative and proapoptotic properties against gastrointestinal cancers [39,40]. Furthermore, *Lactobacillus casei*, *Bifidobacterium longum*, and *L. acidophilus* showed beneficial effects on tumor cell apoptosis [41]. Moreover, in cellular lines, it has been observed that *Bifidobacterium adolescentis* inhibited the proliferation of three human colon cancer cell lines including HT-29, SW 480, and Caco-2 [42]. Studies have shown that at least 107–109 viable bacteria must reach the intestine for health benefits to be achieved for the organism [26,43].

### 3.2. The Role of Microbiota

The gut microbiome maintains a symbiotic relationship with the gut mucosa performing specific metabolic, protective, trophic, and immunomodulatory functions in the organism. Metabolic functions, including production of vitamin K and several components of vitamin B, digestion, and fermentation of the carbohydrates that escaped proximal digestion and indigestible oligosaccharides, result in the synthesis of short-chain fatty acids (SCFA) such as butyrate, propionate, and acetate, which are rich sources of energy for the host. Protective functions are associated with degradation and prevention of the resident pathogen overgrowth, while trophic functions involve control of integrity of the intestinal epithelium and ensure immune system homeostasis [44,45].

The integrity of the intestinal barrier is a hallmark of a eubiotic intestinal ecosystem [33]. Dysbiosis, an imbalance in function or structure of gut microbiota, may be caused by extrinsic factors such as drugs, chemotherapy, radiotherapy, and poor nutrition. Intrinsic factors causing dysbiosis comprise various diseases, such as colitis, inflammatory bowel disease, obesity, and colorectal cancer [46,47].

Ingested probiotic bacteria, which are capable of colonizing the intestinal tract, are reported to restore eubiotic conditions by producing antimicrobial substances such as bacteriocins and lowering the pH in order to inhibit the growth of other pathogenic bacteria [33,48]. Generally, CTx causes a decrease in *Lactobacillus*, *Bifidobacterium*, and other protective bacteria and an increase in specific pathogenic species [49]. In addition, the beneficial probiotic microflora, dominated by *Bifidobacteria* and *Lactobacilli*, are able to modify the gut microbiota by reducing the risk of cancer following their capacity to decrease β-glucoronidase and carcinogen levels [33].

In the GIT, cancer treatment by CTx agents results in intestinal crypt apoptosis and villous atrophy that may affect the composition of luminal microbiota and increase intestinal permeability [50]. Consuming probiotic bacteria can affect the rebuilding of the epithelial barrier by modulating the expression and distribution of tight junction proteins (e.g., occluding, zonula occludens (ZO)-1) [51,52,53]. Both *Bifidobacteria* and *Lactobacilli* increase tight junction protein expression and restore intestinal permeability [54]. Some studies have shown that SCFAs, by activating 5′-adenosine monophosphate-activated protein kinase, a key agent in regulating energy metabolism in colonocytes, leads to a strengthening of the intestinal epithelial tight junctions and creation of a strong and healthy barrier [55].

The epithelial mucus layer is another protective factor; it is regulated by gut bacteria playing an essential role in protecting the host against bacterial invasion and in maintaining the integrity of the intestinal epithelium. Chemotherapy regimens have been shown to alter mucin (MUC) dynamics, potentially reducing intestinal barrier function [56]. Both in vivo and in vitro studies showed the ability of probiotics to increase Muc gene expression and enhance the secretion of mucus by goblet cells [35,57].

Probiotic bacteria may activate cytoprotective pathways in epithelial cells, counteract reactive oxygen species (ROS) displace pathogenic bacteria, interact with tight junctions, and subsequently activate the NF-κB signaling pathway to enhance mucosal integrity and ensure the development of innate immune response. Thus, it contributes to the control of intestinal homeostasis, protection of the gut against injury, promotion of tissue regeneration, maintenance of the barrier function, and eubiotic intestinal microbiota [58,59,60]. 

### 3.3. Pathogenesis of CTx-Induced Mucositis

Almost immediately after initiation of CTx, cellular damage in the intestinal villi becomes evident, whereas clinical evidence of mucositis onset is reported within 24–48 h after treatment start [61,62]. CTx is linked to a range of symptoms such as abdominal pain, diarrhea, constipation, nausea, vomiting, and anorexia. In some cases, dehydration, malnutrition, infections, and sepsis may also occur. These symptoms occur primarily because of direct mucosal damage [63,64].

The pathogenesis of mucositis involves not only the epithelium but also the cells and tissues within the submucosa. Signaling from damaged endothelium, fibroblasts, and infiltrating leukocyte cells contributes to apoptosis, loss of renewal, atrophy, and ulceration. These changes occur slowly in stratified mucosa, whereas in single layers of the small intestine, changes seem to manifest abruptly [65,66].

Animal and human studies revealed mucositis development as a five-step model, entailing complex signaling pathways: (I) An initiation phase with direct DNA injury, the formation of ROS and release of endogenous damage-associated molecular pattern molecules from injured cells of the basal epithelial layers, submucosa, and endothelium. (II) A primary damage response phase with inflammation and apoptosis. This phase starts immediately when DNA strand breaks and the generation of ROS leads to the activation of redox-sensitive transcription factors such as Wnt/β-catenin, p53, caspase-1/3, Bcl-2 and NF-κB, and their associated pathways [67,68,69,70]. The activation of NF-κB leads to the release of proinflammatory cytokines such as tumor necrosis factor (TNF-α), interleukin (IL)-6, IL-1β, IL-6, IL-1, IL-18, IL-33, and cyclooxygenase-2 (COX-2) [63,71,72,73]. The timing of histological lesions, peak tissue levels of NF-κB, and proinflammatory cytokines are different according to the CTx agent (irinotecan, methotrexate, or 5-fluorouracil (5-FU)) [71,72,74]. (III) A signaling and amplification phase, increasing inflammation and apoptosis. For instance, NF-κB activates TNF-α release, which in turn activates more NF-κB. (IV) An ulceration phase, leading to ablation of the epithelial villi, disruption of epithelial cell adhesion, and discontinuity of the epithelial barrier, promoting bacterial translocation and immune cells into lamina propria. (V) A healing phase, with epithelial cell proliferation, migration, and differentiation once chemotherapy or radiotherapy has ceased [12,58,75]. These overlapping steps might be driven by the activation of NF-κB, subsequently promoting key proinflammatory cytokines, causing further mucosal injury, and eliciting further tissue damage [76]. 

Moreover, the small intestine is most often affected. Different CTx agents may target different parts of the cell cycle or metabolism; their effect on intestinal integrity is consistent and characterized by enterocyte hyperplasia, decreased crypt length, blunting and fusion of intestinal villi, and increased apoptosis. Studies with CTx agents suggest that levels of TNFα, IL-1β, and IL-6 are altered in different sites of the alimentary tract prior to histological evidence of damage following CTx [77].

### 3.4. Effects of Probiotics on CTx Side Effects in CRC

The use of probiotics to improve safety and gastrointestinal side effects during cancer treatment has been investigated by evaluating the potential benefits of probiotics during and after CTx. Gastrointestinal toxicity is mainly related to mucosal damage by CTx, decreased colonization resistance, and alteration of the natural host microflora. Probiotics may decrease the risk and severity of CTx-related toxicity, and thus may reduce side effects associated with cancer treatment [78,79].

Probiotics were evaluated mainly in the prevention of infectious complications of CTx, weight loss, and CTx-related diarrhea. In animal models, promising results have been reported (Table 2). Preclinical trials, although using diverging study design, animal populations, and probiotic products, revealed that animals receiving probiotics before, during, and after CTx developed fewer episodes of high-grade diarrhea and proved the safety of use of probiotics.

Studies in mice and rats developing diarrhea following intraperitoneal application of 5-fluorouracil discovered that the symptoms were alleviated after treatment with multistrain probiotics containing *Lactobacillus* and *Bifidobacterium* (*LaBi*). All studies with a *LaBi* mixture showed a protective effect against weight loss compared to the 5-FU group. Average jejunal crypt depth increased significantly returning to near control levels after administration of *LaBi* in the CTx group. Expression of TLR2 and TLR4, TNF-α, IL-1β, IL-4, IL-6, IL-17, and IFNγ in intestinal tissue were significantly reduced after probiotic strains were given to 5-FU-treated mice [76,83,89].

Treating 5-FU-induced side effects with a single probiotic strain in mice and rat studies showed that *B. infantis*, *B. bifidum,* or *L. acidophilus* administration diminished the severity of intestinal damage. This led to reduced MPO activity, TNF-α expression, and IL-1β expression; it also increased GSH and IL10 concentrations, prevented the loss in BW, and reduced the occurrence of diarrhea as well as the decrease in villus height [78,84,88]. The best effect of *B. infantis* was observed at a dose of 10^9^ CFU/mouse. Interestingly, after the first injection of 5-FU, *B. bifidum* failed to prevent the initial induction of apoptosis at 24 h. These findings suggest that *B. bifidum* does not prevent the induction of apoptosis but is able to suppress the secondary inflammatory responses during the progression of 5-FU-induced IM [84]. Moreover, Justino et al. measured gastric emptying and intestinal transit, revealing that *L. acidophilus* reversed 5-FU-induced changes in GIT motility, which enhanced intestinal transit and gastric emptying and decreased retention in the distal bowel segment [78].

Bowen et al. [80] evaluated the multistrain probiotic VSL#3 in the prevention of single intraperitoneal dose irinotecan-induced diarrhea and mucositis. Maximal protective effects of probiotics were achieved when the probiotics were given before and after chemotherapy. VSL#3 significantly reduced intestinal apoptosis, and thus helped to prevent mucosal breakdown and crypt damage. It also increased epithelial proliferation, prevented moderate or severe diarrhea, prevented weight loss, and prevented irinotecan-induced loss in goblet cell numbers.

Another study [81] underlined the activity of *Saccharomyces cerevisiae* in reducing the severity of diarrhea and weight loss in mice after administration of both viable and heat-killed probiotic yeast. Furthermore, only viable probiotic yeast prevented the loss of goblet cells, preserved the architecture of intestinal mucosa, and reduced mucosal inflammation. *S. cerevisiae* decreased oxidative stress induced by irinotecan. Most importantly, intestinal concentration of SN-38 (an active metabolite of irinotecan) remained stable under the yeast treatment, whereas lower intestinal concentrations of active SN-38 could contribute to a decrease of the chemotherapeutic efficacy of irinotecan. Sezer et al. [82] investigated the efficiency of another probiotic from the *Saccharomyces* genus—*Saccharomyces boulardii*—on irinotecan-induced diarrhea and mucosal damage in rats. In rats receiving probiotics, mucosal damage was significantly less and improvement on diarrhea was recorded.

A study by Ching-Wei Chang et al. [19] showed that CTx with FOLFOX is associated with a change in microbial diversity, and oral administration of single strain probiotic *Lactobacillus rhamnosus* (*Lcr35*) restored this compositional change. Their taxonomic analysis indicated that FOLFOX significantly increased the abundance of Firmicutes, decreased the abundance of Bacteroidetes, and increased the F/B (Firmicutes/Bacteroidetes) ratio. Furthermore, *Lcr35* administration restored the crypt depth and alleviated villus height-to-crypt depth ratio in CTx-treated mice, although the levels did not reach those observed in the normal saline group. *Lcr35* administration was able to restore the healthy microbiome as well as reduced the severity of diarrhea and intestinal mucositis by modulation of the proinflammatory responses with suppression of intrinsic apoptosis without affecting the antitumor effect of FOLFOX.

Hui Mi et al. [85] used the CRC rat model and showed that *B. infantis* administration prevented the loss of BW and the decrease in villus height, reduced the occurrence of diarrhea, and reduced the severity of intestinal damage caused by 5-FU and oxaliplatin by suppressing Th1 and Th17 responses.

Not all probiotics ameliorate side effects caused by CTx. Hanru Wang [86], Whitford [87], and Smith CL [11] investigated the effects of *S. thermophilus* and *L. fermentum* in a rat model of CTx-induced mucositis. They showed that *S. thermophilus* and *L. fermentum* only partially prevented the loss of BW and partially reduced jejunal inflammation, but neither treatment was effective at reducing structural and functional changes in the GIT.

## 4. Conclusions

Animal studies showed that use of probiotics may reduce different side effects of CRC CTx treatment including GIT injury, IM, weight loss, and diarrhea. IM is the main side effect after CTx in CRC. The development of mucositis involves changes in gut microbiota and activation of NF-κB. Activated NF-κB results in apoptotic signals and proinflammatory cytokine production, sequentially contributing to GIT injury, diarrhea, and weight loss. Probiotics seem to have potential capacities in prevention of CTx-induced side effects in CRC treatment by modulating the gut microbiota and proinflammatory responses with suppression of intrinsic apoptosis and appear to be a promising alternative therapeutic strategy that targets both the deregulated immune response and the intestinal dysbiosis. Further animal and human studies aiming to investigate the effective dose and combination of different probiotic strains, the effectiveness of probiotics supplementation intervention in reducing inflammatory markers, and the side effects of CTx are required.

## Figures and Tables

**Table 1 ijms-22-09347-t001:** Microorganisms considered as probiotics.

*Lactobacillus* spp.	*Bifidobacterium* spp.	Other Lactic Acid Bacteria	Non Lactic Acid Bacteria
*L. acidophilus*	*B. animalis*	*Streptococcus thermophilus*	*Saccharomyces cerevisiae*
*L. brevis*	*B. adolescentis*	*Enterococcus faecium*	*Saccharomyces boulardii*
*L. casei*	*B. bifidum*	*Pediococcus acidilactici*	
*L. fermentum*	*B. breve*	*Bacillus coagulans*	
*L. johnsonii*	*B. infantis*		
*L. lactis*	*B. lactis*		
*L. paracasei*	*B. longum*		
*L. plantarum*	*B. thermophilum*		
*L. rhamnosus* *L. bulgaricus*			

**Table 2 ijms-22-09347-t002:** Studies conducted on probiotic role in treating and prevention the side effect of CRC treatments in animal studies.

Study	Main Objective	Number and Strain of Animals	CTx Regimen	Probiotics	Major Findings
Chun-Yan Yeung et al. [75]	To investigate the effects and safety of probiotic supplementation in ameliorating 5-FU-induced intestinal mucositis	72 Balb/c mice	A 5-day repeated 30 mg/kg/day intraperitoneal dose of 5-FU	*Lactobacillus casei*, *Lactobacillus rhamnosus*, *Lactobacillus acidophilus*, Bifidobacterium bifidum (1 × 10^7^ cfu/d) daily for 5 days	General condition: in 5-FU + probiotics group the decrease in BW was significantly less severe versus (vs) 5-FU + saline group. Gut function: Diarrhea scores significantly lower after probiotics administration; Histology: 5-FU + probiotics group was significantly increased jejunal villus length, restored crypts depth and increased number of goblet cells vs. 5-FU + saline; Serum analysis: Proinflammatory cytokines TNF-α, IL-1β, IL-6 levels significantly decreased in 5-FU + probiotics group vs. 5-FU + saline;
Joanne M Bowen et al. [77]	To investigate the probiotic mixture, VSL#3, for amelioration of chemotherapy-induced diarrhea	48 female DA rats	A single intraperitoneal dose of 225 mg/kg irinotecan (CPT-11)	VSL3# (3.0 × 10^8^ cfu/d) daily for 21 days pre-treatment and 7 days post-treatment	General condition: Probiotics reduced BW loss; Gut function: Diarrhea scores significantly lower after probiotics administration; Histology: increased crypt proliferation in irinotecan + VSL3# group combined with an inhibition of apoptosis in both the small and large intestines
R.W.Bastos et al. [80]	To evaluate the pre- or post-treatment with viable or inactivated *Saccharomyces cerevisiae* could prevent weight loss and intestinal lesions, and maintain integrity of the mucosal barrier in a mucositis model	88 Swiss male mice	A 3-day repeated 75 mg/kg/day intraperitoneal dose of irinotecan	*Saccharomyces cerevisiae* UFMG A-905 (Sc-905) (1 × 10^9^ cfu/d) daily, 10 days before, during and 2 days after CTx	Only post-treatment with viable Sc-905 was able to protect mice against the damage caused by CTx. General condition: *Saccharomyces cerevisiae* after CTx reduced BW loss. Gut function: yeast reduced intestinal permeability. Histology: Irinotecan + yeast group was significantly increased jejunal villus length, prevented the decrease of goblet cells and stimulated the replication of cells in the intestinal crypts vs. Irinotecan + saline; Oxidative stress assessment: A significant reduce in lipid peroxidation was in Irinotecan + yeast group vs. Irinotecan + saline;
Sezer A et al. [81]	To investigate the efficiency of *Saccharomyces* *boulardii* on irinotecan-induced mucosal damage and diarrhea in rats	50 male Sprague-Dawley rats	A 4-day repeated 60 mg/kg/day intravenously dose of irinotecan	*Saccharomyces boulardii* (800 mg/kg) daily, 3 days before, during and 3 days after CTx	General condition: Probiotics reduced BW loss; Gut function: Diarrhea scores significantly lower after probiotics administration; Histology: Irinotecan + probiotics group was significantly increased jejunal villus length and reduced mucosal edema vs. Irinotecan group;
Ching-Wei Chang et al. [19]	To evaluate the effect of *Lactobacillus casei* variety FOLFOX-induced mucosal injury *rhamnosus* (Lcr35) on	48 BALB/c mice	A 5-day repeated 30 and 10 mg/kg intraperitoneal 5-FU and LV. Single dose of oxaliplatin 1 mg/kg i.p. on first day	*Lactobacillus casei* variety *rhamnosus* Lcr35 (1 × 10^3−7^ cfu/d) daily, 7 days before, during and 2 days after CTx	Gut function: Diarrhea scores significantly lower in FOLFOX + *Lcr35* (1 × 10^7^ CFU/daily) group; Histology: FOLFOX + *Lcr35* (1 × 10^5−7^ CFU/daily) groups was significantly increased jejunal villus length and restored crypts depth vs. FOLFOX group; But FOLFOX + *Lcr35* at the highest dose did not significantly reduce goblet cell damage; Imunohistochemistry: FOLFOX + *Lcr35* (1 × 10^7^ CFU/daily) significantly reduced TUNEL-positive cells, number of p65-reactive cells and BAX-positive cells in the intestine; *Lcr35* did not affect the proliferative activity and caspase-8 protein expression after FOLFOX; *Lcr35* (1 × 10^7^ CFU/daily) significantly suppressed FOLFOX-induced IL-6, TNF-α in jejunum;
Lawrence Huang et al. [82]	To evaluate the safety of probiotic supplementation and to determine the probiotic effect in response to 5-FU intestinal mucositis	36 male SCID/NOD mice	A 5-day repeated 30 mg/kg/day intraperitoneal dose of 5-FU	*Lactobacillus casei* variety *rhamnosus* Lcr35; *Lactobacillus acidophilus*; *Bifidobacterium bifidum LaBi* (1 × 10^7^ cfu/d) daily for 5 days	General condition: in 5-FU + probiotics group the decrease in BW was significantly less severe vs. 5-FU + saline group. *Lac35* had stronger protective effect vs. *LaBi*; Gut function: Diarrhea scores significantly lower after probiotics administration; Histology: 5-FU + probiotics groups was significantly increased jejunal villus length and restored crypts depth vs. 5-FU + saline; Serum analysis: both *Lcr35* and *LaBi* significantly inhibited serum cytokines TNF-α, IL-1β, IFNγ, IL-6, IL-4, IL-10, and IL-17; Lcr35 and LaBi potentially safe therapeutic option with no evidence of bacteremia;
Shinichi Kato et al. [83]	To evaluate the effect of *Bifidobacterium bifidum* on 5-FU-induced intestinal mucositis in mice	35 male mice	A 6-day repeated 50 mg/kg/day intraperitoneal dose of 5-FU	*Bifidobacterium bifidum* G9-1 (BBG9-1) (1 × 10^7−9^ cfu/d) daily for 9 days, begining 3 days before onset of 5-FU	General condition: BW loss was significantly lower in 5-FU + BBG9-1 (1 × 10^9^ CFU/mouse) group; Gut function: Diarrhea scores significantly lower after probiotics administration; Histology: In 5-FU + BBG9-1 (1 × 10^9^ CFU/mouse) group was significantly increased jejunal villus length and restored crypts depth vs. 5-FU group Cytokine and enzyme assessment: MPO, TNF-α and IL-1β levels significantly decreased in 5-FU + BBG9-1 (1 × 10^9^ CFU/mouse) group vs. 5-FU;
Hui Mi et al. [84]	To investigate the effect of *Bifidobacterium infantis* in attenuating the severity of chemotherapy-induced intestinal mucositis in rats with colorectal cancer	30 male Sprague-Dawley rats	Dimethyl hydrazine injected subcutaneously weekly for 10 weeks, and then injected with SW480 cells in rectal mucosa to create a CRC model. On the 8th day, a 3-day repeated 75 mg/kg i.p. of 5-FU and 8 mg/kg i.p. of oxaliplatin	*Bifidobacterium infantis* (1 × 10^9^ cfu/d) daily for 11 days, beginning 8 days before CTx	General condition: Probiotics reduced BW loss; Gut function: Diarrhea scores significantly lower after probiotics administration; Histology: In 5-FU and oxaliplatin + *B. infantis* group was significantly increased jejunal villus length and restored crypts depth vs. 5-FU + saline group; Serum analysis: cytokines TNF-α, IL-1β, L-6 levels were significantly reduced in 5-FU and oxaliplatin + *B. infantis* group; *B. infantis* effectively attenuated chemotherapy-induced intestinal mucositis by decreasing Th1 and Th17 response;
Hanru Wang et al. [85]	To investigate the effects of *Streptococcus thermophilus* in a rat model of mucositis induced by the anthracycline chemotherapy drug, doxorubicin	32 female Dark Agouti rats	A single intraperitoneal dose of 20 mg/kg doxorubicin	*Streptococcus thermophilus* TH-4 (1 × 10^9^ cfu/mL) daily for 9 days, on day 6 received CTx	General condition: TH-4 partially prevented the loss of BW induced by doxorubicin; Histology: TH4 failed to reduce damage of jejunum and ileum tissue: to increase villus length and restore crypts depth after doxorubicin injection; Enzyme assessment: MPO levels significantly decreased in the jejunum in doxorubicin + TH4 group vs. doxorubicin;
Whitford et al. [86]	To investigate *S. thermophilus* (TH-4) for their potential to reduce the severity of 5-FU-induced small intestinal damage in rats	45 female Dark Agouti rats	A single intraperitoneal dose of 150 mg/kg 5-FU	* Streptococcus thermophilus* TH-4 (6 × 10^9^ cfu/mL) live, supernatant and dead formulation daily for 6 days, on day 3 received CTx	General condition: there were no significant differences in reducing BW loss after 5-FU injection + TH-4 live, supernatant or dead formulation; Histology: 5-FU + live and supernatant TH4 significantly reduced crypt fission vs. 5-FU + skim milk; 5-FU + live TH-4 partially normalized mitotic count; Enzyme assessment: no significant difference of MPO levels was in 5-FU + either live, dead or supernatant TH4 group vs. 5-FU + skim milk;
K.-T. Yuan et al. [87]	To evaluate the beneficial effects of *Bifidobacterium infantis* in a rat model of intestinal mucositis induced by 5-fluorouracil	30 male Sprague-Dawley rats	A single intraperitoneal dose of 150 mg/kg 5-FU	*Bifidobacterium infantis* (1 × 10^9^ cfu/d) daily for 11 days, starting from 7 days before CTx	General condition: Probiotics significantly reduced BW loss; Gut function: Diarrhea scores significantly lower after probiotics administration; Histology: In 5-FU + *B. Infantis* group was significantly increased jejunal villus length vs. 5-FU group; Imunohistochemistry: in 5-FU + *B. infantis* group significantly increased expression of proliferating cell nuclear antigen (PCNA), reduced expression of NF-κB vs. 5-FU group; Cytokine and enzyme assessment: plasma cytokines TNF-α, IL-1β and MPO activity were significantly reduced in 5-FU + *B. infantis* vs. 5-FU group;
Yan Tang et al. [88]	To evaluate the effects of a probiotic mixture, DM#1, on intestinal mucositis and dysbiosis of rats treated with 5-fluorouracil	28 male Sprague-Dawley rats	A 5-day repeated 30 mg/kg/day intraperitoneal dose of 5-FU	DM#1 (1 × 10^8−9^) cfu/d) daily, during and 3 days after CTx	General condition: in 5-FU + probiotic group was significantly reduced BW loss vs. 5-FU group; Histology: In 5-FU + DM#1 group was significantly increased ileal villus height and restored crypts depth vs. 5-FU group; Cytokine and enzyme assessment: MPO activity, expression levels of TLR2 and TLR4 and pro-inflammatory cytokines TNF-α, IL-4, IL-6 were significantly reduced in 5-FU + DM#1 vs. 5-FU group; Increased intestinal permeability caused by 5-FU was normalized after administration of DM#1 mixture;
Justino PF et al. [89]	To evaluate the effect of *L. acidophilus* on the inflammatory and functional outcomes of 5-FU-induced IM in mice	24 male Swiss mice	A single intraperitoneal dose of 450 mg/kg 5-FU	*Lactobacillus acidophilus* (16 × 10^9^ cfu/d) daily for 3 days after CTx	General condition: in 5-FU + probiotic group was significantly reduced BW loss vs. 5-FU group; Gut function: slower GI transit, gastric retention and increased retention in the distal bowel segment caused by 5-FU was reversed by treatment with *L. acidophilus*; Histology: In 5-FU + probiotic group was significantly increased ileal and jejunal villus height and restored crypts depth vs. 5-FU group; Cytokine, oxidative stress and enzyme assessment: MPO activity and cytokine TNF-α, IL-1β, CXCL1 levels were significantly reduced in the jejunum and in the ileum in 5-FU + *L. acidophilus* vs. 5-FU group; glutatione (GSH) concentrations and anti-inflammatory cytokine IL-10 level in the jejunum and in the ileum caused by 5-FU was reduced after administration of *L. acidophilus*;
Smith CL et al. [11]	To evaluate *L. fermentum* BR11 potential to decrease the severity of 5-FU-induced small intestinal damage in rats	56 female dark agouti rats	A single intraperitoneal dose of 150 mg/kg 5-FU	*L. fermentum BR11* (1 × 10^6−9^ cfu/d) daily for 9 days starting from 7 days before CTx	General condition: BR11 partially prevented the loss of BW induced by 5-FU; Histology: In 5-FU + probiotic group was no significant differences in ileal and jejunal villus height and crypts depth vs. 5-FU group; Enzyme assessment: MPO activity was significantly reduced in 5-FU + BR11 vs. 5-FU group;

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
