# Peer review of "The Impact of Probiotics on Intestinal Mucositis during Chemotherapy for Colorectal Cancer: A Comprehensive Review of Animal Studies"

_ijms, 2021, doi:10.3390/ijms22179347_

Round 1
Reviewer 1 Report
This is a very well written Review of the current literature regarding probiotics on mucositis during CTX therapy
Some questions are pending:
- Is there a clinical severity grading of mucositis?
- Are there differences in the severity of IM depending on the downstream activated molecules ?
- Do different administration routes of the CTX (oral, iv, intraperitoenal..) resulting in differences in the effect of the different probiotics?
Author Response
Dear reviewer,
Thank You for Your comments. Below You can find answer to Your questions.
- Point 1: Is there a clinical severity grading of mucositis?
- Response 1: Clinical severity of mucositis according to National Cancer Institute-Common Toxicity Criteria is commonly used in clinical practice. However, in the studies included in this review, authors did not grade intestinal mucositis, they did only histological analysis of intestinal mucosa and examined the probiotics effects on the villus height, crypt depth and goblets cells in the jejunum/ileum or/and colonic mucosa. In most cases CTx causes substantial changes in the intestinal mucosal layer including flattened epithelial layer, decreased in goblet cell number, shortened villi length and inflammatory cells infiltration in lamina propria compared to the saline controls. However, in most cases these effects were alleviated significantly following probiotics administrations in mice/rats in CTx groups, though the levels did not reach to that in the normal saline groups.
- Point 2: Are there differences in the severity of IM depending on the downstream activated molecules ?
- Response 2: No difference was observed in researches which was included in this review.
- Point 3: Do different administration routes of the CTX (oral, iv, intraperitoenal..) resulting in differences in the effect of the different probiotics?
- Response 3: Unfortunately, in animals studies we found that only one author in his research (Sezer A. 2008) administrated intravenous CTx and gave Saccharomyces boulardii probiotics so it is difficult to make a comparison. In this research administration of probiotics after i/v CTx showed similar results as i/p CTx: helped to reduce BW loss; diarrhea scores significantly lower after probiotics administration; CTx + probiotics group was significantly increased jejunal villus length and reduced mucosal edema vs CTx group.
Reviewer 2 Report
Considering the title of the review, the theme turns out to be interesting, but I believe that, at present, the authors have not done a very thorough job.
The first part of the review should better introduce the focus topics of the review. Instead, it explains the arguments in a superficial way, with almost no technical or molecular insight to justify what is written. It is difficult to fully understand the focus of the review, subparagraphs 3.1, 3.2, 3.3, 3.4. they are written somewhat superficially. Evidence should be included that justifies and emphasizes what is being reported.
line 16-19. Specify whether these preclinical studies, which have shown these ameliorative effects, have been observed following chemotherapy treatment and if they suggest an actual benefit in preventing or treating therapy-induced damage and symptoms.
lines 45-48. Eliminating the phrase "Some studies have ..." is a repetition. it is already present in paragraph 3.4, where it is better explained.
Lines 97-100: Explain better the beneficial effects of administering probiotics, what effective impact has been observed in the treatment of some pathologies and in the improvement of associated symptoms, what their functional contributions may be and insert the scientific evidence that affirms the use of the probiotics listed in Table 1 in helping to improve the health of the gastrointestinal tract in health and disease conditions.
Section 3.2: Better integrate the role of the intestinal microbiota in conditions of eubiosis and dysbiosis. Insert the importance of having an eubiotic intestinal microbiota and the beneficial and protective effects it induces on the host (eg production of SCFAs, antimicrobial peptides, protection from pathogens, etc.). Explain the role of a dysbiosis, already associated with colorectal cancer, and the negative effects of a dysbiotic condition of the intestinal microbiota for the host's health. I would also underline the importance of the use of probiotics in helping to restore a dysbiotic condition, so as to emphasize the beneficial effects of this supplementation.
Section 3.3: Too synthetic. Integrate and better explain the side effects of CTx, inserting some more evidence in which the damage induced in the intestine by CTx has been observed, or merge paragraph 3.3 and paragraph 3.4 together.
lines 137-153: The molecular mechanisms underlying the development of mucositis could be better described, explaining for example the role of ROS, NF-kB and cell and tissue damage induced by pro-inflammatory cytokines in an individual with colorectal cancer and undergoing with CTx treatment.
Table 2.
- Main objective: Put "Saccharomyces boulardii; Streptococcus thermophilus" in italics.
- Probiotics: Enter, where possible, the dosage of the probiotic, the dosage and the duration of the treatment. Eliminate italics in some inserted dosages.
Paragraph 3.5: OK
Conclusions: The conclusions are too strong and need to be revised. The authors' statements cannot be fully justified by the preclinical data on animal models provided.
It needs to be corrected, deepened and reviewed by the authors.
Author Response
Dear reviewer,
Thank You for Your comments. Below You can find answer to Your questions.
- Point 1: line 16-19. Specify whether these preclinical studies, which have shown these ameliorative effects, have been observed following chemotherapy treatment and if they suggest an actual benefit in preventing or treating therapy-induced damage and symptoms.
- Response 1: Specified information in lines 16-19, that probiotics prevents CTx induced side effects.
- Point 2: lines 45-48. Eliminating the phrase "Some studies have ..." is a repetition. it is already present in paragraph 3.4, where it is better explained.
- Response 2: Phrase eliminated.
- Point 3: Lines 97-100: Explain better the beneficial effects of administering probiotics, what effective impact has been observed in the treatment of some pathologies and in the improvement of associated symptoms, what their functional contributions may be and insert the scientific evidence that affirms the use of the probiotics listed in Table 1 in helping to improve the health of the gastrointestinal tract in health and disease conditions.
- Response 3: Added more information about probiotics (including those from Table 1) beneficial effects in treatment of pathologies such as: ulcerative colitis, Crohn's disease, infections, colon cancer. Lines 110-122
- Point 4: Section 3.2: Better integrate the role of the intestinal microbiota in conditions of eubiosis and dysbiosis. Insert the importance of having an eubiotic intestinal microbiota and the beneficial and protective effects it induces on the host (eg production of SCFAs, antimicrobial peptides, protection from pathogens, etc.). Explain the role of a dysbiosis, already associated with colorectal cancer, and the negative effects of a dysbiotic condition of the intestinal microbiota for the host's health. I would also underline the importance of the use of probiotics in helping to restore a dysbiotic condition, so as to emphasize the beneficial effects of this supplementation.
- Response 4: Added more info about microbiota role in eubosis/dysbiosis, about microbiota importance and how probiotic supplementation helps to keep healthy environment in intestinal tract. Lines 141-180
- Point 5: Section 3.3: Too synthetic. Integrate and better explain the side effects of CTx, inserting some more evidence in which the damage induced in the intestine by CTx has been observed, or merge paragraph 3.3 and paragraph 3.4 together.
- Response 5: Paragraph 3.3 was merged with paragraph 3.4.
- Point 6: lines 137-153: The molecular mechanisms underlying the development of mucositis could be better described, explaining for example the role of ROS, NF-kB and cell and tissue damage induced by pro-inflammatory cytokines in an individual with colorectal cancer and undergoing with CTx treatment.
- Response 6: Added more information about mucositis development and role of ROS, NF-kB, cell damage. Lines 203-218.
- Point 7: Table 2.
- Main objective: Put "Saccharomyces boulardii; Streptococcus thermophilus" in italics.
- Probiotics: Enter, where possible, the dosage of the probiotic, the dosage and the duration of the treatment. Eliminate italics in some inserted dosages. - Response 7: Missing information added. Font also formatted.
- Point 8: Conclusions: The conclusions are too strong and need to be revised. The authors' statements cannot be fully justified by the preclinical data on animal models provided.
- Response 8: Conclusions was revised and edited. Lines 318-329
Reviewer 3 Report
The manuscript entitled “The impact of probiotics on intestinal mucositis during chemotherapy for colorectal cancer. A comprehensive Review of preclinical trials.” submitted for revision in the International Journal of Molecular Sciences had been positively reviewed with some minor modifications.
This is an interesting manuscript on the effects of probiotics on chemotherapy-induced side effects in preclinical studies in the treatment of colorectal cancer and discusses potential mechanisms based on the results of experimental studies.
The review is positive. Therefore, I propose to accept this paper for publication after minor amendments.
- The title of the publication should be changed as the publication is based on the results of animal studies. This information should be added to the title of the publication
- The abstract should include the purpose, e.g.: It may be similar to the Introduction but information on animal studies should also be given in the abstract
- Keywords: should be changed. These words are already in the title of the publication and should not be repeated
- The layout of table 1 should be changed, column 3 is incomprehensible, it suggests that the listed probiotics belong to the group of Streptococcus spp. I believe that table 1 should also include other probiotic microorganisms along with the reference to the publication
- There is a mistake in the title of table 1: not “microorganisms considered as probiotics” or “Microorganisms considered as probiotics”
- The title of Table 2 should be changed as the studies related to animals. This information must be added to the title e. g.: “Studies conducted on probiotic role in treating and preventing the side effect of CRC treatments - iv vivo experiments” or “…in animal studies”
- It should be clearly stated in the Conclusions that this is for animal studies and that the effects should be checked in humans and that the studies should continue
- Authors may also include the publications:
- Marta Molska, Julita Reguła. Potential Mechanisms of Probiotics Action in the Prevention and Treatment of Colorectal Cancer, Nutrients 2019, 11(10), 2453.
- Jahani-Sherafat, S., Alebouyeh, M., Moghim, S., Ahmadi Amoli, H., & Ghasemian-Safaei, H. (2018). Role of gut microbiota in the pathogenesis of colorectal cancer; a review article. Gastroenterology and hepatology from bed to bench, 11(2), 101–109.
- Hofseth, L.J., Hebert, J.R., Chanda, A. et al. Early-onset colorectal cancer: initial clues and current views. Nat Rev Gastroenterol Hepatol (2020). https://doi.org/10.1038/s41575-019-0253-4.
Author Response
Dear reviewer,
Thank You for Your comments. Below You can find answer to Your questions.
- Point 1: The title of the publication should be changed as the publication is based on the results of animal studies. This information should be added to the title of the publication
- Response 1: Changed.
- Point 2: The abstract should include the purpose, e.g.: It may be similar to the Introduction but information on animal studies should also be given in the abstract
- Response 2: added purpose in line 53-55.
- Point 3: Keywords: should be changed. These words are already in the title of the publication and should not be repeated
- Response 3: Changed to gut microbiota, dysbiosis, pro-inflammatory cytokines.
- Point 4: The layout of table 1 should be changed, column 3 is incomprehensible, it suggests that the listed probiotics belong to the group of Streptococcus spp. I believe that table 1 should also include other probiotic microorganisms along with the reference to the publication
- Response 4: Table 1 layout was changed. Added missing probiotic microorganisms from researches included in this review: L. bulgaricus, Saccharomyces boulardii
- Point 5: There is a mistake in the title of table 1: not “microorganisms considered as probiotics” or “Microorganisms considered as probiotics”
- Response 5: Title was corrected.
- Point 6: The title of Table 2 should be changed as the studies related to animals. This information must be added to the title e. g.: “Studies conducted on probiotic role in treating and preventing the side effect of CRC treatments - iv vivo experiments” or “…in animal studies”
- Response 6: The title was corrected to “Studies conducted on probiotic role in treating and prevention the side effect of CRC treatments in animal studies"
- Point 7: It should be clearly stated in the Conclusions that this is for animal studies and that the effects should be checked in humans and that the studies should continue
- Response 7: Conclusions edited, lines 317-320.
- Point 8: Authors may also include the publications:
a) Marta Molska, Julita Reguła. Potential Mechanisms of Probiotics Action in the Prevention and Treatment of Colorectal Cancer, Nutrients 2019, 11(10), 2453.
b) Jahani-Sherafat, S., Alebouyeh, M., Moghim, S., Ahmadi Amoli, H., & Ghasemian-Safaei, H. (2018). Role of gut microbiota in the pathogenesis of colorectal cancer; a review article. Gastroenterology and hepatology from bed to bench, 11(2), 101–109.
c) Hofseth, L.J., Hebert, J.R., Chanda, A. et al. Early-onset colorectal cancer: initial clues and current views. Nat Rev Gastroenterol Hepatol (2020). doi.org/10.1038/s41575-019-0253-4. - Response 8: Publications included. Citation numbers are a) [44] - added more information about eubiosis and dysbiosis in gut microbiota and importance of that. Lines 142-150 b) [47] - added info when wrote about gut microbiota and cancer. Lines 152-156 c) [7] - added more information about factors causing colorectal cancer. Lines 38-40.